# Upregulation of Enhancer of Zeste Homolog 2 (EZH2) with Associated pERK Co-Expression and PRC2 Complex Protein SUZ12 Correlation in Adult T-Cell Leukemia/Lymphoma

**DOI:** 10.3390/cancers16030646

**Published:** 2024-02-02

**Authors:** Jiani Chai, Jui Choudhuri, Jerald Z. Gong, Yanhua Wang, Xuejun Tian

**Affiliations:** 1Montefiore Medical Center, Department of Pathology, Albert Einstein College of Medicine, Bronx, NY 10467, USAywang@montefiore.org (Y.W.); 2Department of Pathology and Genomic Medicine, Thomas Jefferson University Hospital, Philadelphia, PA 19107, USA; jerald.gong@jefferson.edu

**Keywords:** ATLL, T-cell neoplasms, EZH2, H3K27me3, pERK

## Abstract

**Simple Summary:**

Adult T-cell leukemia/lymphoma (ATLL) is a highly aggressive mature T-cell neoplasm with an extremely poor prognosis. For decades, the first-line therapy for ATLL has been CHO(E)P, although treatment refractoriness and relapse are common. Therefore, there is a need to search for new treatment options. This study, focusing on EZH2, an important epigenetic regulator, and associated intracellular signaling molecules, demonstrates that both EZH2 and pERK expression are upregulated in ATLL, contributing to tumor aggressiveness in biopsied patients’ tumor tissues. Their inhibitors could be potential therapeutic targets for these aggressive T-cell neoplasms.

**Abstract:**

EZH2, a subunit of the polycomb repressive complex 2 (PRC2), is an important methyltransferase that catalyzes the trimethylation of histone H3 at lysine 27 (H3K27me3). EZH2 is overexpressed in various malignancies. Here, we investigated EZH2 expression and potential signaling molecules that correlate with EZH2 expression in ATLL and other T-cell neoplasms. Immunohistochemical staining (IHC) was performed for EZH2, pERK, MYC, and pSTAT3 on 43 ATLL cases and 104 cases of other T-cell neoplasms. Further IHC studies were conducted for Ki-67, SUZ12, and H3K27me3 on ATLL cases. All ATLL cases showed EZH2 overexpression. In other T-cell neoplasms, a high prevalence of EZH2 overexpression was identified (86%), except for T-PLL (33%). In ATLL, EZH2 overexpression correlated with pERK co-expression (86%), while only a small subset of cases showed MYC (7%) or pSTAT3 (14%) co-expression. In the other T-cell neoplasms, there was a variable, but higher, co-expression of EZH2 with pERK, MYC, and pSTAT3. In ATLL, enhanced EZH2 expression correlated with higher Ki-67 staining, SUZ12 (another PRC2 subunit), and H3K27me3 co-expression. In conclusion, EZH2 is overexpressed in ATLL and is associated with pERK expression. It correlates with an increased proliferation index, indicating an aggressive clinical course. EZH2 also correlates with SUZ12 and H3K27me3 co-expression, suggesting its PRC2-dependent catalytic activity through trimethylation. Additionally, EZH2 is overexpressed in most T-cell neoplasms, suggesting that EZH2 could function as an oncogenic protein in T-cell tumorigenesis. EZH2 and pERK could serve as potential therapeutic targets for treating aggressive ATLL. EZH2 could also be targeted in other T-cell neoplasms.

## 1. Introduction

Enhancer of zeste homolog 2 (EZH2) is an essential catalytic subunit of the epigenetic regulator polycomb repressive complex 2 (PRC2). PRC2 functions as a histone methyltransferase that trimethylates the 27th lysine residue of histone H3 (H3K27me3), associated with gene expression silencing in cancer [1]. EZH2 is overexpressed in many cancer entities and correlates with a poorer prognosis. Early evidence comes from solid tumor studies where EZH2 overexpression has been observed in multiple cancers, including prostate cancer, breast cancer, bladder cancer, endometrial cancer, and melanoma [2]. More recently, we and other groups found EZH2 overexpression in a range of hematologic malignancies including B-cell lymphomas, Hodgkin lymphomas, histiocytic and dendritic cell neoplasms, subsets of T-cell neoplasms, plasma cell neoplasms, and myeloid neoplasms [3,4,5,6,7,8,9]. Targeting EZH2 has emerged as a new therapeutic approach against the epigenetic mechanisms of carcinogenesis in recent years. There are several ongoing clinical trials using EZH2 inhibitors in different cancer types [10].

Transcriptional activation of EZH2 has been proposed to promote EZH2 overexpression in neoplastic cells. For instance, transcription factors E2F and NF-κB have been shown to induce EZH2 transcription in various cancer cells [11,12]. Signaling pathways, including MEK-ERK1/2, pSTAT3, and MYC signaling were demonstrated to activate EZH2 transcription in different cancer types [13,14,15]. We previously investigated these potential regulators of EZH2 expression in various types of hematologic malignancies and found differential signaling cascades associated with different tumor types. In diffuse large B-cell lymphoma, histiocytic and dendritic cell neoplasms, the pERK signaling cascade significantly correlated with EZH2 expression [4,5]. In Burkitt lymphoma and double hit lymphoma, MYC cascade was associated with EZH2 upregulation [5]. Combined signaling cascades of pERK, MYC, and pSTAT3 were involved in EZH2 expression in nodular lymphocyte-predominant Hodgkin lymphoma (NLPHL), classic Hodgkin lymphoma (cHL), T-cell/histiocyte-rich large B-cell lymphoma (THRLBCL), and B-cell Lymphoma, unclassifiable, with features intermediate between diffuse large B-cell lymphomas and classic Hodgkin lymphomas (BCLu-DLBCL/cHL) [6]. These regulators of EZH2 expression may serve as additional therapeutic targets.

Adult T-cell leukemia/lymphoma (ATLL) is a rare and highly aggressive mature T-cell neoplasm with an extremely poor prognosis. ATLL is linked to the human T-cell lymphotropic virus type 1 (HTLV-1) infection. For decades, the first-line therapy for ATLL has been CHO(E)P (cyclophosphamide, doxorubicin, vincristine, (etoposide), and prednisone), although refractoriness and relapse are common. Recent advances in research on the epigenetic regulation of ATLL have suggested EZH2 as a promising target for treatment. EZH2 expression was significantly upregulated in primary ATLL cells. This upregulation contributed to miRNA silencing and NF-kB activation, leading to apoptosis resistance and tumorigenesis [16]. The epigenomic and transcriptomic analysis of primary ATLL cells revealed excessive H3K27me3 accumulation associated with EZH2 upregulation. This leads to ATLL-specific abnormal gene downregulation that was also detected at an early stage of disease progression. Furthermore, EZH2 inhibition reversed the ATLL epigenetic disruption and selectively eliminated leukemic- and HTLV-1-infected cells [11]. In case studies, EZH2 overexpression has been reported in ATLL and a range of T-cell neoplasms [9,17,18]. However, the reported case numbers have been limited so far. Here, we investigated EZH2 expression in ATLL biopsy specimens and compared this with other T-cell neoplasms, including T-lymphoblastic leukemia/lymphoma (T-ALL), anaplastic large cell lymphoma (ALCL), anaplastic lymphoma kinase (ALK)-positive (ALCL-ALK+), ALCL, ALK-negative (ALCL-ALK−), extranodal NK/T-cell lymphoma, nasal type (NK/TCL), peripheral T-cell lymphoma not otherwise specified (PTCL-NOS), angioimmunoblastic T-cell lymphoma (AITL), and T-cell prolymphocytic leukemia (T-PLL). In addition, to understand the signaling pathway involved in EZH2 activation in ATLL and other T-cell neoplasms, we studied the intracellular signaling cascade-associated molecules, pERK1/2, CMYC, and pSTAT3, in relation to EZH2 expression.

## 2. Materials and Methods

### 2.1. Patient Selection and Regulatory Approval

Following hospital institutional review board approval, we retrospectively reviewed medical records to identify ATLL cases and other types of T-cell lymphomas, including T-ALL, ALCL-ALK+, ALCL-ALK−, NK/TCL, PTCL-NOS, AITL, and T-PLL. These diagnoses were established according to the 2017 revised 4th edition WHO classification of Hematopoietic and Lymphoid Tissues. A representative block of each case with diagnostic tissue was selected for immunohistochemical (IHC) stains.

### 2.2. Immunohistochemistry

Formalin-fixed, paraffin-embedded tissue blocks were cut into 4 μm-thick sections using a standard technique. IHC staining was performed via a standard protocol using the Dako automated immunostainer (Agilent, Santa Clara, CA, USA). Heat-induced epitope retrieval was completed using Bond Epitope Retrieval Solution 2 (catalog#AR9640), which is an EDTA-based pH 9.0 solution, for 20 min at 100 °C. Primary antibodies used include the following: EZH2 (Cell Signaling Technology, Danvers, MA, USA; catalog #5246), p53 (DAKO, Carpinteria, CA, USA; catalog# GA61661-2), pSTAT3 (Tyr705) (Cell Signaling Technology, Danvers, MA, USA; catalog #9145), cMYC (Abcam, Cambridge, UK; catalog #ab32072), pERK1/2 (Thr202/Tyr204) (Cell Signaling Technology, Danvers, MA, USA; catalog #4370), Ki-67 (Abcam, Cambridge, UK; catalog #ab16667), SUZ12 (Abcam, Cambridge, UK; catalog #ab126577), and H3K27me3 (Cell Signaling Technology, Danvers, MA, USA; catalog #9733). Primary antibodies were applied at the optimized dilution for 30 min at ambient temperature. Staining was performed using the Bond Polymer Refine Detection Kit DS9800 (Leica Biosystem, Wetzlar, Germany) according to the manufacturer’s guidelines. Tissue sections were counterstained with hematoxylin. The cases were scored for the percentage of positive tumor cells (0% to 100%) and for staining intensity (0 to 3) in the representative visual fields. Based on our previous publication [6], EZH2 staining was considered to be overexpressed if ≥60% of the neoplastic cells exhibited 2+ or 3+ staining intensity. PERK, pSTAT3, and MYC staining were considered positive if ≥5% of the neoplastic cells displayed 2+ or 3+ staining intensity. Cases were evaluated and scored by two hematopathologists, respectively.

### 2.3. Statistical Analysis

Statistical analysis was performed by GraphPad Prism, Version 8 (San Diego, CA, USA). Pearson’s correlation between two variables in serial sections was used to analyze the correlation between EZH2 expression and Ki-67/SUZ12. Pearson’s correlation coefficient (r) was used to measure the linear correlation between two variables. Differences were considered significant at *p* < 0.05.

## 3. Results

### 3.1. ATLL Patient Demographics and Survival Data

We retrieved clinical information for 36 of the 43 ATLL cases. Seven cases were diagnosed outside the hospital, with incomplete clinical data. Among the 36 ATLL cases, the average age of these patients was 60 years (ranging from 25 to 83 years). There were 20 female patients (55.6%) and 16 males (44.4%). Most patients were from the Caribbean area (30/36; 83.3%), one patient was from West Africa (1/36; 2.8%), and five patients had unknown endemic origin (5/36; 13.9%). Among different variants of ATLL, acute and lymphomatous variants are most common in our cohort (acute: 11/36; 30.5%; lymphomatous: 10/36; 27.8%). One patient was classified as having a chronic variant (1/36; 2.8%), and one patient was classified as having a smoldering variant (1/36; 2.8%). There was not enough clinical data to classify the remaining 13 patients. Most of the patients were deceased at the time of the study (24/36; 66.7%), with an average survival time of 57.3 months (ranging from 0.1 to 721.6 months). The demographic and survival data are summarized in Table 1.

### 3.2. EZH2 Is Overexpressed in ATLL and Correlates with an Increased Tumor Proliferation Index

Our study included a cohort of 43 ATLL cases with an aggressive clinical course. We assessed EZH2 expression in the ATLL specimens using IHC staining. Using a 60% cutoff for EZH2 overexpression, as suggested by previous studies [6], all the cases (100%) in our cohort overexpressed EZH2 (Figure 1A,B and Table 2). The EZH2 positive staining intensity ranged from 2+ to 3+ nuclear staining. We also performed P53 IHC staining as a surrogate to assess the underlying TP53 mutation status in our cohort. TP53 is the most commonly mutated gene in human cancers and is associated with adverse prognoses in many cancers. However, in our cohort, all the cases were negative for strong uniform P53 expression (<20% positivity) (Figure 1C), suggestive of a TP53 wild-type pattern [19].

EZH2 has been shown to promote tumor cell proliferation and is associated with a high proliferation index in many types of cancer, including breast, prostate, endometrium, and melanoma [20]. We examined the tumor cell proliferation in ATLL by assessing Ki-67 nuclear staining, and correlated Ki-67 staining with EZH2 expression. We found a significant positive correlation between EZH2 expression and Ki-67 staining (r = 0.5467, *p* < 0.05) (Figure 1D,E), suggesting that EZH2 upregulation in ATLL is associated with an enhanced tumor proliferation index.

### 3.3. EZH2 Expression Is Upregulated in a Range of T-Cell Neoplasms in Addition to ATLL

In the other T-cell neoplasms we examined, a high prevalence of EZH2 upregulation was detected in most of the T-cell neoplasms except in T-PLL (Table 2). Similar to ATLL, all the T-ALL (19/19, 100%) and ALCL-ALK− (12/12, 100%) cases showed EZH2 overexpression. In ALCL-ALK+, NK/TCL, PTCL-NOS, and AITL, EZH2 overexpression was detected in the majority of cases: ALCL-ALK+ (13/14, 93%), NK/TCL (15/16, 94%), PTCL-NOS (15/16, 94%), and AITL (15/17, 88%). Different from the other types of T-cell neoplasms, in T-PLL, only a small percentage of cases showed EZH2 overexpression (3/9, 33%). Representatives of EZH2 staining in each type of T-cell neoplasm are shown in Figure 2.

### 3.4. EZH2 Upregulation Is Associated with pERK in ATLL and Correlates with Different Signaling Molecules in Other T-Cell Neoplasms

Signaling cascade molecules, including pERK, pSTAT3, and MYC have all been implicated in promoting EZH2 transcriptional activation in different cancers. Whether these mechanisms apply in ATLL or other T-cell neoplasms is unclear. We then investigated EZH2 expression in association with MYC, pSTAT3, and pERK. In ATLL, most cases (86%) co-expressed pERK, whereas only a small subset of cases showed MYC (7%) or pSTAT3 (14%) co-expression. This indicates that the pERK signaling pathway could be involved in EZH2 upregulation in the majority of ATLL patients.

In the other T-cell lymphomas, there was variable co-expression of EZH2 with pERK, MYC, and pSTAT3. In ALCL-ALK+, NK/TCL, and PTCL-NOS, co-expression of pERK (62%, 73%, 47%), MYC (92%, 67%, 67%), and pSTAT3 (77%, 73%, 47%) was detected. In ALCL-ALK−, a high percentage of cases co-expressed pERK (58%) and MYC (82%), and only a small subset of cases showed PSTAT3 co-expression (33%). In AITL, a high percentage of cases co-expressed pERK (85%) and pSTAT3 (54%). Therefore, it is possible that two or three of these signaling pathways could be associated with EZH2 upregulation in these T-cell neoplasms. In contrast, T-ALL showed a low frequency of EZH2 co-expression with pERK, MYC, or pSTAT3. Thus, it is likely that none of these pathways significantly contributed to EZH2 upregulation in T-ALL. The results are summarized in Table 2. Representative images of pERK, pSTAT3, and MYC staining in each type of T-cell neoplasm are shown in Figure 3.

### 3.5. Association of EZH2 Overexpression with PRC2 Complex Protein SUZ12

The oncogenic mechanisms of EZH2 in hematologic malignancies are complicated and depend on the tumor type. In addition to the canonical histone methyltransferase activity, EZH2 also plays a role in PRC2-independent transcription activation [10]. To investigate whether EZH2 upregulation in ATLL is associated with an enhanced H3K27 histone methyltransferase activity, we assessed H3K27me3 in our cohort and found that all ATLL cases showed positive H3K27me3 (Figure 4A).

Next, we studied the association of EZH2 expression with the PRC2 complex protein SUZ12, which is another member of the polycomb protein family, together with EZH2 and other proteins, which forms the PRC2 complex. In our cohort of ATLL, SUZ12 expression showed a positive correlation with EZH2 (r = 0.4013, *p* < 0.05) (Figure 4B,C). Therefore, EZH2 overexpression in ATLL is associated with an increased histone methyltransferase activity, which could contribute to oncogenesis through gene silencing.

## 4. Discussion

Here, we analyzed EZH2 expression and its association with tumor proliferation and signaling cascade molecules in 43 ATLL cases and a total of 103 cases of other types of T-cell lymphomas, including T-ALL, ALCL-ALK+, ALCL-ALK−, NK/TCL, PTCL-NOS, AITL, and T-PLL. To our knowledge, this is the largest cohort studied for EZH2 expression and regulation in T-cell neoplasms using in vivo patient samples. In this study, EZH2 was overexpressed in all ATLL cases and in the majority of other T-cell neoplasms we examined, except T-PLL. Our findings are consistent with previous cohort studies that reported EZH2 overexpression in the T-cell neoplasms [9,17,18]. The widespread overexpression of EZH2 in these T-cell neoplasms suggests that EZH2 could function as an oncogenic protein in T-cell tumorigenesis.

One important mechanism to promote EZH2 overexpression in neoplastic cells is through the transcriptional activation of EZH2. The ERK, MYC, and STAT3 signaling pathways have all been implicated in the activation of EZH2 in different types of cancer, including hematopoietic malignancies [4,5,6,9,13,14,15]. In T-cell neoplasms, a previous study showed that EZH2 expression was correlated with the expression of MYC and/or pSTAT3 in a subset of T-cell neoplasms. However, the mechanism by which EZH2 expression is mediated in ATLL was unclear. Here, we showed that pERK, MYC, and pSTAT3 were differentially expressed in different T-cell neoplasms. Uniquely in ATLL, EZH2 overexpression was associated with pERK co-expression. MYC/pSTAT3 were detected in only a small subset of ATLL cases. In AITL, both ERK and pSTAT3 were correlated with EZH2 expression. In ALCL-ALK−, EZH2 expression was associated with ERK and MYC co-expression. All three of these molecules were potentially involved in EZH2 regulation in ALCL-ALK+, PTCL-NOS, and NK/TCL. In T-ALL, none of these mechanisms seemed to be involved in EZH2 regulation. A proposed model of EZH2 overexpression in association with different intracellular signaling pathways in these T-cell neoplasms is shown at the in vivo protein expression and activation levels in Figure 5.

EZH2 overexpression and its correlation with pERK in ATLL tissues suggests that pERK-signaling-mediated EZH2 upregulation could contribute to tumorigenesis in ATLL. Establishing a causal relationship is a challenge for this retrospective study. Further studies addressing the functional significance of altered EZH2/pERK expression in tumor progression would be necessary to strengthen this hypothesis. The role of EZH2 in carcinogenesis involves cell cycle regulation. It has been shown that in melanoma, breast, prostate, and endometrium cancer, upregulation of EZH2 is associated with a high proliferation index [20]. In primary ATLL cells, EZH2 upregulation contributes to miRNA silencing and NF-kB activation, leading to apoptosis resistance and tumorigenesis [16]. Furthermore, EZH2 inhibition reverses the ATLL epigenetic disruption and selectively eliminates leukemic- and HTLV-1-infected cells [11]. Consistent with these findings, we show a significant correlation between EZH2 expression and Ki-67 levels in ATLL specimens, supporting the potential role of EZH2 in regulating tumor cell proliferation by these in vitro studies.

The oncogenic mechanisms of EZH2 in hematologic malignancies are complicated and depend on the tumor type. In diffuse large B-cell lymphoma, EZH2 is involved in tumor suppressor gene silencing through its H3K27 histone methyltransferase activity. Inhibition of EZH2 methyltransferase activity results in reduced H3K27 methylation and activation of PRC2 target genes, which have been shown to inhibit proliferation and induce cell cycle arrest [21]. In addition to its canonical histone methyltransferase activity, EZH2 also functions in a PRC2-independent manner. In mantle cell lymphoma, EZH2 expression shows no correlation with H3K27me3 and weakly correlates with the other PRC2 complex components EED and SUZ12 expression, suggesting that EZH2 might have a PRC2-independent role in mantle cell lymphoma [8]. In ATLL, we found that EZH2 expression correlated with H3K27me3 and PRC2 complex component SUZ12 expression. Therefore, EZH2 upregulation in ATLL is associated with enhanced histone methyltransferase activity.

Targeting EZH2 emerges as an important strategy for cancer treatment and shows promising results. Several EZH2 inhibitors have been developed and used in clinical trials in recent years. These molecules selectively block EZH2 methyltransferase activity and reduce global H3K27 methylation [21,22]. Additionally, EZH1, a homolog of EZH2 in the non-canonical PRC2 complex, functions as a methyltransferase and complements EZH2 in mediating H3K27 methylation [23]. Dual EZH1/EZH2 inhibitors have also been developed and demonstrated an enhanced efficacy than targeting EZH2 alone both in vitro and in vivo [24,25,26]. In 2020, tazemetostat, a selective EZH2 inhibitor, was approved by the US Food and Drug Administration (FDA) to treat patients with epithelioid sarcoma and refractory follicular lymphoma [27]. Compared with solid tumors and B-cell lymphomas, targeting EZH2 in T-cell lymphomas is still in the early stages of investigation. In a preclinical study with a patient-derived xenograft model of hepatosplenic T-cell lymphoma (HSTL), treatment with tazemetostat significantly prolonged the survival of mice with this aggressive lymphoma [28]. In a recent phase 2 clinical trial (www.clinicaltrials.gov, NCT04102150, accessed on 30 January 2024), a dual EZH2 and EZH1 inhibitor, valemetostat, was used to treat patients with relapsed or refractory (R/R) ATLL. This study demonstrated promising efficacy and tolerability of valemetostat in heavily pre-treated patients [29].

## 5. Conclusions

ATLL is a highly aggressive T-cell neoplasm with an extremely poor prognosis. The median survival for the acute and lymphomatous variants is estimated to be 8 to 10 months [30]. For decades, the first-line therapy for ATLL has been CHO(E)P, although treatment refractoriness and relapse are common. Therefore, there is a need to search for new treatment options. Our study demonstrated that EZH2 could be a potential therapeutic target for ATLL and some other T-cell neoplasms. An early phase clinical study showed promising results of EZH inhibition in treating ATLL patients [29], warranting further investigation in this field.

## Figures and Tables

**Figure 1 cancers-16-00646-f001:**
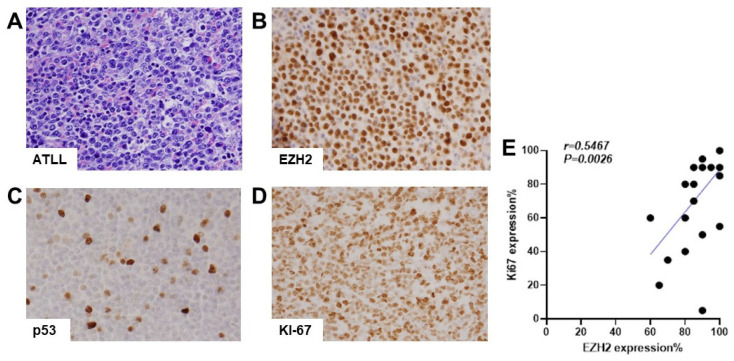
EZH2 is overexpressed in ATLL and correlates with an increased tumor proliferation index. (**A**–**D**) Representative images of H&E staining (**A**) and IHC staining for EZH2 (**B**), p53 (**C**), and Ki-67 (**D**) in one ATLL case (400×). (**E**) Pearson’s correlation of EZH2 expression with Ki-67 expression (r = 0.5467, *p* < 0.05).

**Figure 2 cancers-16-00646-f002:**
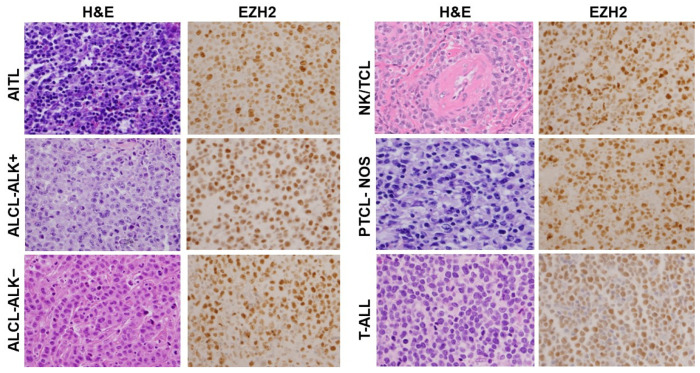
EZH2 overexpression is detected in a wide range of T-cell neoplasms. Representative images of H&E staining and EZH2 IHC staining in AITL, ALCL-ALK+, ALCL-ALK−, NK/TCL, PTCL-NOS, and T-ALL are shown (400×). Abbreviations: AITL: Angioimmunoblastic T-cell lymphoma; ALCL-ALK+: Anaplastic large cell lymphoma, ALK positive; ALCL-ALK−: Anaplastic large cell lymphoma, ALK negative; NK/TCL: Extranodal NK/T-cell lymphoma, nasal type; PTCL-NOS: Peripheral T-cell lymphoma, NOS; T-ALL: T-lymphoblastic leukemia/lymphoma.

**Figure 3 cancers-16-00646-f003:**
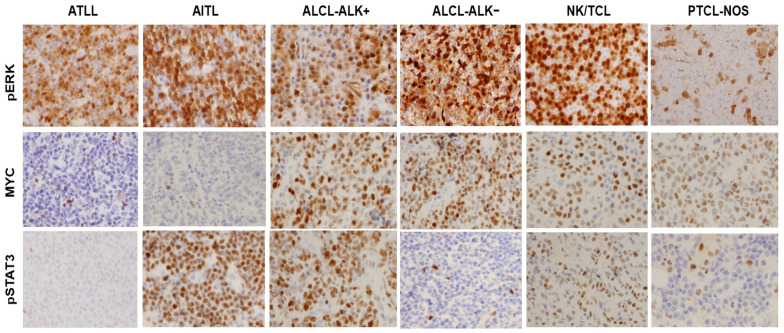
EZH2 upregulation is associated with pERK in ATLL and correlates with different signaling molecules in other T-cell neoplasms. Representative IHC images for pERK, MYC, and pSTAT3 staining in T-cell neoplasms with EZH2 co-expression, including ATLL, AITL, ALCL-ALK+, ALCL-ALK−, NK/TCL, and PTCL-NOS (400×).

**Figure 4 cancers-16-00646-f004:**
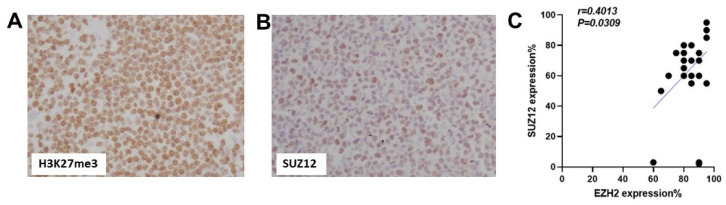
EZH2 upregulation in ATLL is associated with an enhanced histone methyltransferase activity. (**A**) Representative image of IHC stain for H3K27me3 in ATLL cases (400×). (**B**) Representative image of IHC stain for SUZ12 in ATLL cases (400×). (**C**) Pearson’s correlation of EZH2 expression with SUZ12 expression (r = 0.4013, *p* < 0.05).

**Figure 5 cancers-16-00646-f005:**
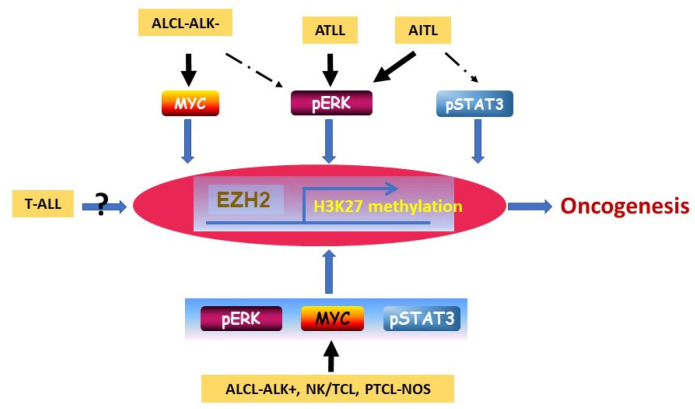
Proposed model of EZH2 overexpression in association with different intracellular signaling molecules pERK, pSTAT3, and MYC that contribute to tumorigenesis in various types of T-cell neoplasms. In ATLL, upregulation of EZH2 with associated pERK co-expression and enhanced PRC2-dependent histone methyltransferase activity that contribute to oncogenesis.

**Table 1 cancers-16-00646-t001:** Demographics and Survival Data of the ATLL Cohort.

Variables	N = 36
Age at diagnosis, mean ± SD	60.0 ± 14.5
(Minimum–Maximum)	(25–83)
Sex, *n* (%)	
Male	16 (44.4)
Female	20 (55.6)
Endemic origin, *n* (%)	
Caribbean	30 (83.3)
West Africa	1 (2.8)
Not available	5 (13.9)
Variants, *n* (%)	
Acute	11 (30.5)
Lymphomatous	10 (27.8)
Chronic	1 (2.8)
Smoldering	1 (2.8)
Not available	13 (36.1)
Survival data, *n* (%)	
Alive	12 (33.3)
Deceased	24 (66.7)
Survival months, mean ± SD	57.3 (142.8)
(Minimum–Maximum)	0.1–721.6

**Table 2 cancers-16-00646-t002:** EZH2 expression and association with pERK, MYC, and pSTAT3 expression in ATLL and other T-cell neoplasms.

T-Cell Neoplasms	EZH2 (POS/Total)	pERK	MYC (POS/Total)	pSTAT3
ATLL	43/43 (100%)	37/43 (86%)	3/43 (7%)	5/43 (11%)
AITL	15/17 (88%)	13/15 (85%)	4/15 (27%)	8/15 (54%)
ALCL-ALK+	13/14 (93%)	8/13(62%)	12/13 (92%)	10/13(77%)
ALCL-ALK−	12/12 (100%)	7/12 (58%)	9/11 (82%)	4/12 (33%)
NK/TCL	15/16 (94%)	11/15 (73%)	10/15 (67%)	11/15 (73%)
PTCL-NOS	15/16 (94%)	7/15 (47%)	10/15 (67%)	7/15 (47%)
T-ALL	19/19 (100%)	0/18 (0%)	5/17 (30%)	1/16 (6%)
T-PLL	3/9 (33%)	0/3 (0%)	1/3 (33%)	0/3 (0%)

Abbreviations: POS: positive; ATLL: Adult T-cell leukemia/lymphoma; AITL: Angioimmunoblastic T-cell lymphoma; ALCL-ALK+: Anaplastic large cell lymphoma, ALK positive; ALCL-ALK−: Anaplastic large cell lymphoma, ALK negative; NK/TCL: Extranodal NK/T-cell lymphoma, nasal type; PTCL-NOS: Peripheral T-cell lymphoma, NOS; T-ALL: T-lymphoblastic leukemia/lymphoma; T-PLL: T-cell prolymphocytic leukemia.

## Data Availability

Data are contained within the article.

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
