# Peer review of "Upregulation of Enhancer of Zeste Homolog 2 (EZH2) with Associated pERK Co-Expression and PRC2 Complex Protein SUZ12 Correlation in Adult T-Cell Leukemia/Lymphoma"

_cancers, 2024, doi:10.3390/cancers16030646_

Round 1
Reviewer 1 Report
Comments and Suggestions for Authors
The investigation by Chai et al, suggested in the manuscript that EZH2 is consistently overexpressed in ATLL and exhibits high prevalence in various T-cell neoplasms. This overexpression is associated with pERK expression, indicating potential therapeutic targets. Notably, in ATLL, EZH2 correlates with increased proliferation, suggesting an aggressive clinical course, and co-expression with SUZ12 and H3K27me3 implies PRC2-dependent catalytic activity. The authors suggested EZH2 and pERK could serve as potential therapeutic targets for aggressive ATLL, and EZH2 is a target in other T-cell neoplasms. There are some major concerns that need to address to support the conclusion.
Major concerns:
ü Consider exploring variations in EZH2 and its correlation with signaling pathways across different gender or age groups. This additional analysis could enhance the study's impact, offering a comprehensive understanding of potential differences in EZH2 expression and signaling pathways among diverse demographics.
ü Strengthen the discussion by incorporating specific examples or referencing clinical studies that demonstrate the efficacy of EZH2 inhibitors in T-cell neoplasms. This addition would enhance the credibility of the proposed approach.
ü The authors need to provide more specific examples or elaborate on the types of T-cell neoplasms, other than ATLL, where targeting EZH2 could be considered a potential therapeutic approach? This would improve the comprehension of the broader applications of EZH2 as a therapeutic target in various T-cell neoplasms.
ü The author needs to thoroughly address potential limitations or challenges of the study.
Comments on the Quality of English Language
Moderate editing of English language required
Reviewer 2 Report
Comments and Suggestions for Authors
The manuscript presents consistent data demonstrating the correlation of EZH2 expression with pERK and the PRC2 complex protein SUZ12 in "aggressive" leukemia (ATLL).
Major concerns:
1. Title: The title employs the abbreviation PRC2, while the text uses the full term "PRC2 complex protein SUZ12." It's important to confirm whether this is intentional or an oversight.
2. Phosphorylated forms of proteins: For pSTAT and pETK1/2, it would be valuable to include precise information about the sites of phosphorylation for a more detailed understanding.
3. Pearson Correlation: Clarify the methodology for determining Pearson correlation. Provide details, such as whether it involves the correlation between two proteins in serial sections or within the same patient. Confirm if it is not a direct colocalization of two proteins in the same tissue section.
4. Section 2.2: In this section, offer detailed data that enables the reproducibility of results related to the antigen retrieval method. Include information on the buffer, temperature, treatment time, and/or any steps involving microwave treatment.
Minor points:
1. Line 22: Ensure uniformity in the use of the abbreviation for pERK (versus p-ERK). Check the text to maintain consistency.
2. Table 2: Clarify the meaning of the abbreviation POS; does it denote "positive"?
3. R-value in Pearson Correlation: If applicable, consider specifying whether the R-value should be R2, especially if it pertains to fitting the curve with linear regression.
Addressing these points will enhance the clarity and precision of the manuscript, ensuring a more thorough understanding for readers and facilitating the potential reproduction of the results.
Reviewer 3 Report
Comments and Suggestions for Authors
In this manuscript, the authors have evaluated the expression of EZH2 and other signaling molecules (pERK, MYC, pSTAT3) in Adult T-cell leukemia/lymphoma (ATLL) and other T cell neoplasms. They observed high expression of EZH2 in all ATLL cases along with H3K27me3, and differential expression of other signaling molecules. In addition, expression of EZH2 correlates with proliferation of ATLL cases. Based on these results, the authors hypothesize that EZH2 could be a potential therapeutic target for ATLL and additional T-cell neoplasms.
Although the data presented is adequate to ascertain expression of EZH2 in ATLL, the functional consequence of EZH2 in relation to other signaling molecules is unclear. The title suggests that upregulation of EZH2/pERK etc contributes to tumor aggressiveness yet there is no loss of function/gain of function data in the manuscript that demonstrates role of either in tumor progression. Given prior studies cited within the manuscript, further exploration of EZH2/pERK cooperativity would elevate the novelty of the study. The authors evaluate expression of some signaling molecules; given that prior studies have posited NFkB to increase EZH2, were NFkB pathway proteins also increased in all ATLL cases? What was the status of EZH1? The authors might also want to cite studies that have evaluated EZH2/EZH1 inhibitors (https://doi.org/10.1182/blood.2022016862) as a therapeutic strategy and discuss additional synergistic pathways.
Minor point: Further clarification regarding acute/chronic status of ATLL, when tissues were obtained would be useful.
Round 2
Reviewer 3 Report
Comments and Suggestions for Authors
The authors have added key points to the discussion describing other EZH2 inhibitors and addressing limitations of the study, thus overall adequately addressing this reviewer's comments.